# Terbium-Based AGuIX-Design Nanoparticle to Mediate X-ray-Induced Photodynamic Therapy

**DOI:** 10.3390/ph14050396

**Published:** 2021-04-22

**Authors:** Joël Daouk, Mathilde Iltis, Batoul Dhaini, Denise Béchet, Philippe Arnoux, Paul Rocchi, Alain Delconte, Benoît Habermeyer, François Lux, Céline Frochot, Olivier Tillement, Muriel Barberi-Heyob, Hervé Schohn

**Affiliations:** 1Department of Biology, Signals and Systems in Cancer and Neuroscience, UMR 7039 Research Center for Automatic Control (CRAN), Université de Lorraine–French National Scientific Research Center (CNRS), F-54000 Nancy, France; joel.daouk@univ-lorraine.fr (J.D.); mathilde.iltis@yahoo.fr (M.I.); denisebechet@hotmail.com (D.B.); alain.delconte@univ-lorraine.fr (A.D.); herve.schohn@univ-lorraine.fr (H.S.); 2Reactions and Chemical Engineering Laboratory (LRGP), UMR 7274, Université de Lorraine–French National Scientific Research Center (CNRS), F-54000 Nancy, France; batoul.dhaini@univ-lorraine.fr (B.D.); philippe.arnoux@univ-lorraine.fr (P.A.); celine.frochot@univ-lorraine.fr (C.F.); 3Light Matter Institute, UMR-5306, Université de Lyon–French National Scientific Research Center (CNRS), F-69000 Lyon, France; paul.rocchi@univ-lyon1.fr (P.R.); francois.lux@univ-lyon1.fr (F.L.); olivier.tillement@univ-lyon1.fr (O.T.); 4Porphychem SAS, F-21000 Dijon, France; b.habermeyer@porphychem.com

**Keywords:** glioblastoma multiforme, AGuIX^®^, terbium, gadolinium, photodynamic therapy, X-ray-induced photodynamic therapy, singlet oxygen

## Abstract

X-ray-induced photodynamic therapy is based on the energy transfer from a nanoscintillator to a photosensitizer molecule, whose activation leads to singlet oxygen and radical species generation, triggering cancer cells to cell death. Herein, we synthesized ultra-small nanoparticle chelated with Terbium (Tb) as a nanoscintillator and 5-(4-carboxyphenyl succinimide ester)-10,15,20-triphenyl porphyrin (P1) as a photosensitizer (AGuIX@Tb-P1). The synthesis was based on the AGuIX@ platform design. AGuIX@Tb-P1 was characterised for its photo-physical and physico-chemical properties. The effect of the nanoparticles was studied using human glioblastoma U-251 MG cells and was compared to treatment with AGuIX@ nanoparticles doped with Gadolinium (Gd) and P1 (AGuIX@Gd-P1). We demonstrated that the AGuIX@Tb-P1 design was consistent with X-ray photon energy transfer from Terbium to P1. Both nanoparticles had similar dark cytotoxicity and they were absorbed in a similar rate within the cells. Pre-treated cells exposure to X-rays was related to reactive species production. Using clonogenic assays, establishment of survival curves allowed discrimination of the impact of radiation treatment from X-ray-induced photodynamic effect. We showed that cell growth arrest was increased (35%-increase) when cells were treated with AGuIX@Tb-P1 compared to the nanoparticle doped with Gd.

## 1. Introduction

Glioblastoma multiforme (GBM) is one of the main incurable brain tumors, mainly due to the presence of infiltrated cells within the parenchyma, responsible for GBM recurrence into the surrounding brain tissue [1]. The conventional treatment of GBM tumors consists of surgical resection followed by X-ray radiation and adjuvant temozolomide administration which improves modestly patient survival [2]. Brain exposure to X-ray involves the generation of oxidative stress, which are responsible for DNA alteration, lipid peroxidation, protein oxidation, and cell redox statue changes, triggering cells to cell death [3]. However, these effects are not limited to malignant cells, but also alter surrounding cells.

Alternative therapeutic strategies have been developed, notably photodynamic therapy (PDT). PDT appears as an innovative technology being investigated to fulfil the need for a targeted cancer treatment that may reduce recurrence and extend survival with minimal side effects [3,4]. It aims at selectively killing neoplastic cells by the combined action of a photosensitizer and visible light in the presence of oxygen, whose combined action mainly results in the formation of reactive species, especially singlet oxygen which is the main mediator of PDT reaction. To improve PDT efficiency, photosensitizer can be bound to ligands such as monoclonal antibodies or peptide moieties and be delivered by carrier systems such as nanoparticle [5,6]. Moreover, the nanoparticles can be modified by functional groups for additional biochemical properties. In addition, nanoparticles accumulation in the solid tumor site is improved by the enhanced permeability and retention effect (EPR) [7,8]. Numerous clinical studies, including phase III randomized prospective clinical trials, have been reported for PDT, using alternative methods such as interstitial PDT and intraoperative PDT [9,10,11,12,13,14,15,16,17,18,19,20]. Interstitial PDT offers a localized treatment approach in which improvements in local control of GBM may result in significant enhanced survival [11,12,20]. Several photosensitizers have been used, including porfimer sodium (Photofrin^®^), 5-aminolevulinic acid (5-ALA, Gliolan^®^), m-tetrahydroxyphenylchlorin (mTHPC, temoporfin, Foscan^®^) and benzoporphyrin derivative monoacids ring A (BPD-MA, verteporfin, Visudine^®^).

Compared to radiotherapy, the light irradiation used in PDT is less energetic and it cannot penetrate deeply enough into the tumor, as most tissue chromophores absorb visible light commonly used in clinical practice [4,6]. The penetration depth of 630 nm light in brain-adjacent-to-tumor is estimated at 2.5 mm. A breakthrough strategy to treat GBM via nanomedicine and X-ray has been suggested by combining the principles of radiotherapy and PDT, both clinically proven modalities, while maintaining their main benefits and decreasing their drawbacks. The principle of the so-named X-ray-induced PDT (X-PDT) is based on the conversion of X-ray photons into visible photons, known as X-ray excited optical luminescence, from the nanoscintillator embedded in the nanoparticle and linked to the photosensitizer, which, in turn, produces singlet oxygen and other oxygen reactive species [21,22]. X-PDT proof-of-concept with nanoparticles was first introduced by Cheng and Wang, who described simultaneous radiation and X-ray-induced photodynamic effects [23]. The strategy requires nanoparticles, exhibiting appropriated physical properties to establish energy transduction from the nanoscintillator to the photosensitizer, a high scintillation quantum yield and an optimal energy transfer from the scintillator onto the photosensitizer [24,25]. It must be pointed that only PDT can generate singlet oxygen which is highly cytotoxic to tumor tissue and to treat deeply lesions without invasive approach such as interstitial PDT. It is possible to use X-ray as an excitation source instead of light. Thus, the light penetration problem through the tumor tissue is overcome, and the activation of the photosensitizer within tumor tissue is performed by classical radiotherapy using X-ray. In addition, the cumulative effects between conventional radiotherapy and PDT should allow the use of conventional X-ray doses. In metal-hybrid system, the metal-based nanoparticle consists of a nanoscintillator coated with polyethylene glycol or a polysiloxane layers to ensure biocompatibility which allows covalently coupling of the photosensitizer [24,25]. Members of the lanthanide family have been used in nanoparticle synthesis, such as mesoporous lanthanum fluoride doped with Cerium or Terbium (Tb) and grafted with porphyrin derivative, Tb_2_O_3_ coated with a polysiloxane layer or silica-doped with lanthamide [26,27,28,29,30,31].

Among them, ultra-small Gadolinium (Gd) based nanoparticles, namely AGuIX^®^, were developed [32]. The nanoparticle design was first proposed for a non-toxic resonance magnetic agent and its imaging properties [33]. Moreover, in vitro and in vivo pre-clinic experiments demonstrated that AGuIX doped with Gd act as a theranostic agent, enhancing radiosensitization of tumor cells in diverse experimental conditions, notably, at different photon radiation energies (with a range from kiloelectron volts to million electron volts) and with different types of radiation [34]. The radiosensitizing effects are associated to diverse processes: Gd mediated generation of electrophotons and Auger photons amplifying the local production of reactive oxygen derived species, as demonstrated for Gold embedded nanoparticles [35,36]; or an impairment of DNA breaks reparation, as a consequence of irradiation, and reactive species production triggering cells to cell death [34,37]. In X-PDT, the main goal of the treatment consists of the transfer energy from irradiated nanoscintillator to the photosensitizer, limiting the delivery of high radiation energies and deposits to kill cancer cells without any alteration to adjoining normal cells. Recently, we demonstrated that AGuIX@ doped with Gd and 5-(4-carboxyphenyl succinimide ester)-10,15,20-triphenylporphyrin (P1) can be used to target Neuropilin-1, a transmembrane receptor localized in endothelial cells within mouse grafted human GBM tumors [38]. The synthesized nanoparticle, referred as AGuIX@Gd-P1, behaved with similar properties to the original AGuIX@Gd.

Therefore, in order to use the AGuIX platform for X-PDT, we suggested the replacement of Gd in the Gd-based AGuIX nanoparticle by Tb as a nanoscintillator and the grafting of P1 (Scheme 1). In these conditions, an external light source will not be required to simultaneously support a photodynamic effect. The nanoparticles (referred herein as AGuIX@Tb and AGuIX@Tb-P1) were characterised for their photophysical and chemical properties. We evaluated the effect of X-PDT on human GBM U-251 MG cell survival after cell exposure to nanoparticles. In parallel, we tested AGuIX@ doped with Gd and P1 (referred herein as AGuIX@Gd and AGuIX@Gd-P1), which has been characterised previously [38]. We highlighted that chelated Tb-P1 nanoparticles can react linearly to X-ray energy and flow, and were able to activate P1 to produce singlet oxygen. In vitro, using human U-251 MG glioblastoma cells, experiments confirmed the interest of these AGuIX design, notably at a 3.0 Gy.min^−1^ dose rate. Moreover, cell exposure to AGuIX@Tb-P1 improved the effect on cell growth arrest when it was compared to similar treatment with AGuIX@Gd or AGuIX@Gd-P1.

## 2. Results

### 2.1. Characteristics of the AGuIX@Tb-P1

We already demonstrated that the grafting of P1 on AGuIX doped with Gd induced a hydrodynamic diameter at about 11.1 nm, twice as large as the original AGuIX@Gd nanoparticle, with an estimated diameter at 4.9 nm on average [38]. Replacing Gd of the original AGuIX-designed nanoparticle by Tb did not induce any size modification. The hydrodynamic diameters of AGuIX@Tb and AGuIX@Tb-P1 were estimated, respectively, at 3.8 ± 1.0 nm and 11 ± 0.8 nm. Moreover, the ζ potential raised from −11.8 to −45.6 mV, when measurements were achieved with AGuIX@Tb and AGuIX@Tb-P1, respectively, supporting a high stability of the latter conjugate (Appendix A). AGuIX@Tb and AGuIX@Gd emission spectra after UV light excitation are presented in Figure 1 as well as P1 absorption spectrum, in water. An overlay between P1 absorption spectrum and Tb emission (Figure 1a,b) could be observed. In contrast, Gd emission did not overlay P1 emission spectrum. Calculation of the spectral overlap between Tb emission and P1 absorption was estimated at J = 1.15 × 10^14^ M^−1^ nm^4^ cm^−1^. The corresponding Förster radius was found to be 2.5 nm (from 1 up to 10 nm). Moreover, Tb luminescence lifetime was 390 µs, long enough to allow energy transfer to P1. Thus, Tb luminescence in presence of P1 showed an exponential decay of its fluorescence intensity (Figure 1c), and a linear decrease of its fluorescence lifetime at PS concentration higher than 0.5 µM (Figure 1d). We recorded the luminescence exponential decay of AGuIX@Tb and AGuIX@Tb-P1 (Appendix A). AGuIX@Tb fluorescence lifetime was estimated at 1 ms whereas AGuIX@Tb-P1 fluorescence lifetime was 1 µs. This decrease of fluorescence lifetime of Tb in presence of P1 supported the concept of energy transfer between Tb and P1. Moreover, we measured the luminescence of both nanoparticles after excitation at 351 nm with a 50 µs delay between excitation and photon detection at 545 nm. As shown in Appendix A, P1 luminescence was obtained between 630 and 690 nm, corresponding to the energy transfer between Tb and P1, concomitantly to the decrease of Tb emission. Collectively, the results obtained allowed us to conclude that Tb energy transfer to P1 is a FRET/non radiative transfer type characterised by a quenching constant, Kq = 0.045 × 10^9^ M^−1^ s^−1^.

### 2.2. Nanoscintillator Response to X-ray Excitation

AGuIX@Tb emission spectrum after X-ray excitation presented a similar profile compared with UV/visible light excitation (Figure 1b). To validate the pipeline acquisition setup under X-ray sessions, X-ray spectroscopy experiments were performed at different tube currents and voltages. Spectra intensities were positively related to tube current from 0.5 to 12.5 mA at 320 kVp (Figure 2a) and to tube voltage from 25 to 320 kVp at 12.5 mA (Figure 2b). For each condition, the four characteristic emission peaks of Tb cations (Tb^3+^) were detected at 485, 545, 590, and 620 nm, respectively. For these different Tb cation emission peaks, the correlation coefficients between tube currents and peak intensities were 0.97, 0.99, 0.99, and 0.84, respectively (Figure 2c). The maximum peak values were associated linearly with the tube voltage values (Figure 2d). Correlation coefficient values were found to be 0.96, 0.99, 0.99, and 0.96 for the four Tb peaks 485, 545, 590, and 620 nm, respectively.

### 2.3. Energy Transfer and Singlet Oxygen Production

We highlighted reactive species and singlet oxygen production using fluorescent probes, respectively, SOSG (Figure 3a) and APF (Figure 3b), during X-ray exposure. X-ray parameters were set to 320 kVp and 12.5 mA as it provided the highest Tb scintillation. Both APF and SOSG signals increased continuously during X-ray exposure. Addition of sodium azide (NaN_3_), a singlet oxygen quencher, confirmed that the type II-PDT reaction (singlet oxygen generation) was mainly involved in X-PDT as SOSG and APF fluorescence signals were mostly inhibited.

### 2.4. Cytotoxicity of AGuIX@Tb-P1 on Glioblastoma Cell Growth

We assessed whether treatment of P1 alone or AGuIX@Tb led to U-251 MG cytotoxicity, using the MTT procedure. Since P1 is hydrophobic, we chose ZnPy3P1 which is soluble in culture medium. IC_50_ was estimated at 34.8 ± 9.9 µM after 24 h and 10.4 ± 3.4 µM after 72 h-treatment duration. In addition, IC_50_ was similar when GBM cells were exposed to AGuIX@Tb with IC_50_ estimated at 1.73 ± 0.3 and 1.56 ± 0.1 mM, after 24 and 72 h exposition, respectively. We finally tested the effect t of increasing concentration (1.0 to 20.0 µM; concentrations are expressed as P1 equivalent throughout the paper) of AGuIX@Tb-P1 and AGuIX@Gd-P1 (Appendix A). No cytotoxicity was observed whatever the dose tested and treatment duration.

### 2.5. NPs Cell Uptake

Cell uptake kinetics were established after cell exposure to 2.5 and 5 µM AGuIX@Tb-P1 over 48 h (Figure 4a). Cell uptake was quantified based on the fluorescence emission of P1. NPs accumulated within the cells, reaching a maximum at 48 h. Since the AGuIX doped with Gd or Tb are synthesized with a similar scheme and behave similar hydrodynamic diameter, we compared NP uptake after cell exposure to 1.0 or 2.5 µM AGuIX@Tb-P1 or AGuIX@Gd-P1 for 24 h. There was no significant difference suggesting that the replacement of Gd by Tb did not affect NP absorption within the cells (Figure 4b). Moreover, when cells were treated with 1 µM AGuIX@-complexes for 24 h (Figure 4b), uptake was low as compared to the results of cell absorption performed with 2.5 or 5.0 µM nanoparticles for the same culture delay (Figure 4a). Since NP absorption could lead to the increase of oxidative stress within the cells, we assessed whether nanoparticle uptake was related to reactive species generation, using DCF2-DA probe. A slight but significant increase was observed after cell exposure to 2.5 µM nanoparticle. However, cell exposure to 1 µM AGuIX@Gd-P1 (Figure 4c) was enough to modify redox statue within the cells. We evaluated whether AGuIX@Tb-P1 uptake is associated with stress-mediated cell death by quantifying propidium iodide positive cells, after cell exposure to 2.5 or 5 µM over 48 h. No change was observed, supporting that the accumulation of AGuIX@Tb-P1 was not related to cell death (Figure 4d).

### 2.6. Photodynamic Effect on U-251 MG Cell Survival

We assessed whether cell exposure to photodynamic treatment led to cell growth inhibition after 24 h exposure duration and could limit cell clone formation in anchorage-dependent clonogenic assays. U-251 MG cells were pre-treated with 1.0 and 2.5 µM AGuIX@Tb-P1 and exposed to a red light (630 nm, 0.7 W, irradiance at 4.54 mW.cm^−2^) to a fluence range of 2.5 to 10.0 J cm^−2^, corresponding to an exposition duration of 2 min to 38 min). At the dose of 1 µM, cell growth inhibition was obtained with a fluence applied at 10.0 J cm^−2^ (Figure 5a). We observed that clone formation was inhibited when the cells were pre-treated with 1 µM AGuIX@Tb-P1 and exposed to a fluence at 2.5 (50%-inhibition) or 5.0 J cm^−2^ (90%-inhibition) (Figure 5b). Similarly, cell growth inhibition was observed in pre-treated U-251 MG cells with 2.5 µM nanoparticle and a fluence at 2.5 J cm^−2^ (20%-inhibition). Cell growth arrest increased with fluence up to 10.0 J cm^−2^ (80% inhibition) (Figure 5a). Since PDT is associated with the generation of oxidative stress, we quantified reactive species content immediately after cell exposure to red light (Figure 5c), when cells were pre-treated with 2.5 µM AGuIX@Tb-P1 and exposed to a fluence at 2.5 J cm^−2^, which corresponds to an immediate cell growth inhibition estimated at 20%. We used DCF2-DA probes which reacts with several reactive oxygen-derived species and gives a valuable estimation of the oxidative stress generated within the cells. We found that reactive species content was enhanced 2 times over 30 min post light exposition, supporting the concept that cell survival depends mainly on oxidative stress-mediated by light treatment. Finally, at nanoparticle concentrations higher than 2.5 µM (data not shown), and a fluence higher than 2.5 J cm^−2^ (Figure 5b), the applied treatment killed all the cells.

### 2.7. X-ray-Induced Photodynamic Effect on U-251 MG Cell Survival

Similarly, we studied whether X-ray irradiation promoted U-251 MG cell growth arrest by photodynamic-mediated effect. Reactive species generation was quantified immediately after cell exposure to X-ray irradiation (Figure 6a). X-ray ionization in itself generated reactive species and Tb scintillation could be involved in the redox change within the cells [5]. We assessed whether oxidative stress was generated in AGuIX@Tb and AGuIX@Tb-P1 pre-treated cells and irradiated at 2.0 Gy, with an energy set at 320 kVp, over 1 h post irradiation (Figure 6a), using DCF2-DA probe. Reactive species content was increased up to 1.5 time when cells were exposed to AGuIXTb-P1, whatever the delay. Conversely, at 30 min post exposition, reactive species content was enhanced in AGuIX@Tb pre-treated cells, but the increase was not significant when the results were compared to those obtained with untreated cells. Moreover, the obtained levels in pre-treated and untreated cells were close, after 1 h post irradiation.

In order to validate the concept of a photodynamic effect in X-PDT strategy, we performed anchorage-dependent clonogenic assays with AGuIX@NPs. Because Gd and Tb are neighbors in terms of atomic number, their behavior regarding X-ray interaction is considered as similar. Cells were pre-treated with 1 µM AGuIX@NPs and irradiated at X-ray doses ranging from 0.5 up to 5.0 Gy, either at 160 or 320 kVp, corresponding to the dose rate of 1.5 or 3.0 Gy.min^−1^, respectively. These conditions were chosen since cell clone formation was mainly inhibited when cells were pre-treated with 2.5 µM AGuIX@Tb-P1 and irradiated for X-ray doses higher than 2.0 Gy at 320 kVp (Figure 6b). Experimental results were plotted in the quadratic semi log model (Figure 7a) and compared (Figure 7b). Curve parameters were computed for survival factor at 2.0 Gy (Table 1). Because we used X-ray dose exposures up to 5.0 Gy, which were close to those classically used in clinical practice (1.5 to 2.5 Gy), we considered the β-parameter equal to 0. In fact, β -parameter governs the slope for high dose exposure (sub-lethal damage), while the α-parameter reflects the enhanced benefit of X-ray-induced PDT on cell survival id est the direct lethal cell damage [39]. As shown in Table 1 and Figure 7b, significant survival difference (*p* = 0.0018) was found at 2.0 Gy between scintillating (AGuIX@Tb-P1) and non-scintillating (AGuIX@Gd-P1) nanoparticles, when X-ray energies were set to 320 kVp corresponding to a mean 116 keV X-ray energy; DMF was estimated of 0.650 versus 0.876. The DER at 2.0 Gy was estimated at 1.54. Finally, for the same Gy dose, the enhanced factor was 35% when cells were treated with AGuIX@Tb-P1. Moreover, we did not find any significant change between results from cells treated with AguIX@Gd-P1 and those obtained of the cells treated with AguIX@Gd for the same experimental conditions (Figure 7b). At energies set at 160 kVp, we did not find any difference between both nanoparticles either doped with Tb or Gd with or without P1. Accordingly, our results support the concept of AGuIX@Tb-P1 photodynamic-mediated effect, whereas nanoparticle chelated with Gd did not.

## 3. Discussion

### 3.1. Gadolinium Substitution by Terbium in the AguIX@ Platform

X-ray-induced PDT represents an alternative to PDT leading to the possibility to access to tumors localized in deep brain tissue. As shown herein, we proposed to replace Gd in the original AGuIX@Gd by Tb and grafted P1 to demonstrate the interest of such nanoparticle in X-PDT. This strategy is based on several lines of evidence: as demonstrated previously, AGuIX@Gd-P1 accumulate within human cell-grafted tumor by EPR in orthotopic site [38,40], even though the nanoparticle delivery depends mainly on the development of angiogenesis processes. Moreover, both AGuIX@Gd and AGuIX@Gd-P1 are eliminated from the animal body by the renal route [34,38]. However, in the latter, nanoparticle clearance was associated to the liver/feces body elimination. Moreover, we showed that the replacement of Gd by Tb, in addition to the grafting of P1, did not change the hydrodynamic diameter of the nano-objects.

Tb was selected as a scintillating agent and we demonstrated the spectral overlap between Tb and P1, which is necessary for X-PDT (Figure 1 and Figure 2). Thus, Tb can transfer the energy received after nanoparticle X-ray irradiation to P1 by FRET non radiative transfer. These results are consistent with previous findings, which highlighted the relation between the 545 nm Tb emission peak, P1 Q3 band and X-ray photons ranging from 20 to 130 kVp in activating photo-agents, through induced visible luminescence from rare-earth particles [31]. In PDT, a large part of visible light is absorbed by the photosensitizer which produces singlet oxygen, known as the PDT type II reaction [3]. In contrast, when X-PDT is used, only a small fraction of the X-ray emitted photons will be converted into scintillations [21,22]. Indeed, the interaction between material and high energy photons depends on the Z atomic number id est electronic density and incident energy [4,21]. Usually, in medical imaging, scintillation crystals are designed thick enough to increase the probability to completely stop incident photons via the photoelectric effect [41]. Conversely, in aqueous media, the effective density will be lower than expected, leading only to a small fraction of X-ray energy converted into visible light. Moreover, the photodynamic efficiency can depend on various factors such as the distribution of the scintillating nanoparticles within the tumor tissue, the composition of the tumor stromal microenvironment, the level of molecular oxygen in this microenvironment, and the illumination density resulting from the scintillating agents. Bulin et al. and Abliz et al. demonstrated the production of singlet oxygen via porphyrin X-ray-induced activation of Gd oxysulfide doped with Tb or Tb oxide, respectively [27,30]. However, low X-ray energies from 10 to 130 kVp (id est mean energies ranging from 5 to 70 keV at 20 mA) were tested, while energies used in radiotherapy are commonly from 160 to few hundred keV for preclinical in vivo experiments. In addition, clinical radiotherapy involves much higher energies, around 6 MV. Scintillation yield is dramatically lower at such levels of keV energies since the probability of photoelectric interaction becomes minimal [41]. Therefore, X-PDT under conventional linear accelerators should be less effective than with radiograph tubes. On the other hand, the bioluminescence of Tb as a scintillating agent was demonstrated, using high concentrations of the nanoscintillator and irradiation deposits without compatibility with the energies in pre-clinical studies or in the clinic [42,43,44].

### 3.2. Irradiation of AGuIX@Tb-P1 Produces Singlet Oxygen

As shown in Figure 3, AGuIX@Tb-P1 produced singlet oxygen under X-ray irradiation, as demonstrated with APF and SOSG probes. APF probe reacts with the hydroxyl radical and singlet oxygen; SOSG is specific to singlet oxygen [45,46,47,48]. Recently, it has been reported that the SOSG fluorescent signal occurs independently either of the presence of singlet oxygen or in the absence of photosensitizer during X-ray irradiation [49]. It has been also shown that the probe under UV excitation, generated an endoperoxide derivative which acts as a photosensitizer producing singlet oxygen [50]. No dramatic change was observed when the SOSG probe was tested without nanoparticles (Figure 3a), at a dose rate of 3.0 Gy.min^−1^ and with an energy set up at 320 kVp. When sodium azide, as a quencher of singlet oxygen, was added in the reaction mixture, the changes observed were mainly due to the production of singlet oxygen generation from P1. Alternatively, the APF probe was used instead of SOSG in the same experimental conditions (Figure 3b). In the experimental conditions used, we could not exclude that during X-ray irradiation of AGuIX@Tb-P1, the APF fluorescence signal increase was associated to the generation of reactive species (especially hydroxyl radical), as observed by the slight but continuous increase of the slope curve, even though in the presence of sodium azide (Figure 3a). Such an increase has been related to X-ray-mediated water radiolysis alone in the presence of the APF probe [51]. However, Bulin et al. [27] hypothesized that P1 acts as a radiosensitizer and therefore the molecule could involve the generation of reactive species under X-ray exposition.

### 3.3. Irradiation of AGuIX@Tb-P1 Induces a Photodynamic Effect on U-251 MG Cell Growth

We assessed whether AGuIX@Tb-P1 altered U-251 MG cell survival by PDT and X-PDT. We used U-251 MG cells whose behavior is close to that of GBM in situ [52]. Collectively, we demonstrated that P1 grafted into AGuIX@Tb limits the cytotoxicity of the molecule alone (Appendix A), in agreement with previous findings [53,54]. AGuIX@Tb-P1 pre-treated cell exposure to photodynamic or X-ray irradiation triggered cell death and limited cell clone formation (Figure 6b and Figure 7). Interestingly, we found that the treatment efficacy was enhanced with X-PDT relative to radiation treatment when cells were exposed to AguIX@Tb-P1 instead of AguIX@Gd-P1 at 2.0 Gy (320 kVp), with a DMF estimated at 0.65 corresponding to an enhanced factor calculated at 35%. The benefit of X-PDT relative to radiation treatment was obtained with diverse nanoparticle designs, for examples, with P1 and Ce-doped with titanium oxide on A549 human lung cancer cell; with P1-grafted silicium oxide nanospheres on Hela cervical cells; with rat L9 glioma cell exposed to LaF_3_:Tb particles with adsorbed meso-tetra(4-carboxyphenyl)porphyrin with low irradiation source; or with SrAl_2_O_4_:Eu^2+^ nanoparticles; collectively, allowing to the conclusion that tumors cells undergo cell death by cumulative/synergistic effects of irradiation treatment and X-ray induced photo-treatment [55,56,57,58]. Finally, cancer cell exposure to AGuIX doped with Gd involved cell radiosensitization. However, we did not find any significant change when cells were treated with AGuIX@Gd, as compared to the results of untreated cells exposed to X-rays alone (Figure 7 and Table 1). This discrepancy could be explained since the radiosensitizing effects on cancer cells from diverse origin has been reported for Gd concentration from 100 µM up to 1 mM [34], whereas herein the concentration of lanthamide was 16 and 25 µM for Tb and Gd, respectively; in both cases, the P1 equivalent concentration tested was 1 µM.

In conclusion, we showed that replacement of Gd by Tb in the initial AGuIX design leads to a promising nanoparticle for X-PDT in in vivo experimental. Moreover, we demonstrated that chelated Tb nanoparticles react linearly to X-ray energy (at least up-to 320 kVp) and flow, and they were able to activate P1 to produce singlet oxygen. The constructed nanoparticle was not toxic, while remaining unexposed to light. In vitro experiments confirmed the interest of these AGuIX design, notably at a 3.0 Gy.min^−1^ dose rate. However, it has to be noted that such a strategy should not be considered with external radiotherapy but with low-energy devices (corresponding to a few hundred of keV), such as radiograph tubes, to ensure high energy conversion and singlet oxygen production while lowering exposure dose. Such investigations are currently being conducted in our group.

## 4. Materials and Methods

### 4.1. Reagents

Fluorescent probes, 3′-(p-aminophenyl) fluorescein (APF), 2′,5′(di-acetate) dichlorofluorescein (DCF2-DA), Singlet Oxygen Sensor Green™ (SOSG) and propidium iodide were from Molecular Probe (Merck-Sigma, St Quentin Fallavier, France). 3-(4,5-dimethylthiazol-2-yl)-2,5-diphenyl tetrazolium (MTT) was purchased from Acros Organics (Thermo Fisher Scientific, France). 5-(4-carboxyphenyl succinimide ester)-10,15,20-triphenylporphyrin (P1) and zinc(II) 5-(4-carboxyphenyl)-10,15,20-(tri-N-methyl-4-pyridyl) porphyrin trichloride (ZnPy3P1) were purchased from Porphychem (Porphychem SAS, Dijon, France). Other reagents were of analytical grade.

### 4.2. Synthesis and Preparation of the AGuIX@-Complexes

We used four different nanoparticles, based on the same AGuIX^®^ polysiloxane core surrounded by 1,4,7,10-tetraazacyclododecane-1,4,7,10-tetraacetic acid (DOTA)/metal cation (3+) complexes, covalently grafted to the inorganic matrix. The cations used were Terbium (Tb^3+^, Z = 65, A = 159 g mol^−1^) and Gadolinium (Gd^3+^, Z = 64, A = 157 g mol^−1^). P1 was covalently grafted as a photosensitizer (AGuIX@Tb-P1 or AGuIX@Gd-P1).

Ultra-small siloxane particles were obtained in a two-step synthesis procedure as described previously [33]. In brief, 3-aminopropyl triethoxysilane (185 mmol) and 1,4,7,10-tetra-azacyclododecane-1-glutaric anhydride-4,7,10-triacetic acid (137 mmol, DOTAGA) were mixed to react in 630 mL of diethylene glycol (DEG) at room temperature for 20 h in order to create the DOTAGA silane. Tetraethyl orthosilicate (249 mmol) was added and the mixture was left for 1 h with stirring before addition of 6.3 L of ultrapure water for the condensation Sol-Gel reaction and hydrolysis. The mixture was heated successively at 80 °C for 6 h, then at 50 °C for 18 h. After incubation, the pH was adjusted to 2.0 with HCl (12N). The ultra-small siloxane particles were purified by tangential filtration on Vivaflow^®^ membranes with a cut-off at 5 kDa (Sartorius Stedim Biotech, Aubagne, France). The final volume was 400 mL with the purification factor of 1000. The pH of the solution was adjusted to 7.4 with 1 M NaOH solution. The final amount of free DOTA was measured by europium titration, as described previously [31]. Free DOTA groups were estimated at 100 mM DOTA. Tb chelation was performed by addition of 1.5 mmole trichloroterbium hexahydrate in 15 mL of the particle solution. The pH was adjusted at 6.0 with 1 M NaOH, and the mixture heated until temperature raised 80 °C. Each day, the pH was measured and adjusted to 6.0, until there was no pH change. The ultra-small particles were filtered onto VivaSpin^®^ membranes (cut-off at 5 kDa, Sartorius Stedim Biotech). The volume obtained was 15 mL and the purification factor was 100. The Tb chelation yield was determined by Inductively Coupled Plasma–Mass Spectroscopy and was estimated at 60%. Finally, the pH was adjusted at 7.4, before freeze-drying.

P1 was grafted following the protocol described previously [33,39]. In summary, 500 mM (1.5 g) of DOTA-free particles, obtained from the first step of particle synthesis, were dispersed in 3 mL of pure water and 1.5 mmol of trichloroterbium hexahydrate. The mixture’s pH was adjusted to 6.0 with 1 M NaOH solution and heated to 80 °C for Tb chelation as described above. 12 mL of DEG pre-heated at 40 °C was added to the solution. 150 µmol P1 diluted in DMSO was added drop-by-drop under stirring. The final solvent ratio H_2_O/DEG/DMSO was 13/53/34. The mixture was left stirring at 40 °C for 12 h in the dark and was filtrated through VivaSpin^®^ membranes (cut-off at 5 kDa, Sartorius Stedim Biotech). The purification factor was 100 and the final volume was 15 mL. The Tb chelating yield, determined by Inductively Coupled Plasma–Mass Spectroscopy, was 47%. The P1 coupling yield was estimated at 29% by recording the absorbance at 520, 550 nm, 590 and 650 nm Q bands. The final Tb:P1 ratio was 16 moles Tb for 1 mole P1. All concentrations of nanoparticle containing P1 will be referred to thereafter as the concentration of P1. A stock AGuIX@Tb-P1 suspension was prepared in water and was 3 mM P1 equivalent; the solution of AGuIX@Tb was 37.5 mM (Tb equivalent) in water.

### 4.3. Dynamic Light Scattering Size

Direct measurements of the size distribution of the nanoparticles suspended in any medium were performed via Zetasizer NanoS DLS (Dynamic light scattering, laser He-Ne 633 nm, Malvern Instrument, Orsay, France). Prior to the experiment, the nanoparticles were diluted in 0.01 M NaCl (pH 7.4).

### 4.4. Potential of the AGuIX Conjugates

Potential ζ measurements were carried out with a Zetasizer Nano-Z (Malvern Instrument) equipped with a He-Ne laser at 633 nm. Before measurement, the nanoparticles were dispersed in 0.01 M NaCl and buffer solutions (Honeywell Fluka^TM^ Buffer Solution, ThermoFischer Scientific, Ilkirch, France).

### 4.5. Synthesis of AGuIX@Gd and AGuIX@Gd-P1

The AGuIX@ complexes were synthesized and characterized as described previously [33], with a ratio of 1 mole of P1 for 25 moles of Gadolinium (Gd), as estimated by Inductively Coupled Plasma–Mass Spectroscopy analysis. Stock solution of AGuX@Gd or AguX@Gd-P1 was respectively 50 mM Tb equivalent, and 4 mM P1 equivalent in water.

### 4.6. Photophysical Properties of AGuIX@Tb-P1

Absorption spectra were recorded on a UV-3600 UV-visible double beam spectrophotometer (Shimadzu, Marne-La-Vallée, France). Fluorescence spectra were recorded on a Fluorolog FL3-222 spectrofluorometer (Horiba Jobin Yvon, Longjumeau, France) equipped with a 450 W Xenon lamp and thermostated cell compartment (25 °C), a UV-visible photomultiplier R928 (Hamamatsu Photonics, Hamamatsu, Japan), and an InGaAs infrared detector (DSS-16A020L Electro-Optical System Inc., Phoenixville, PA, USA). The excitation beam was diffracted by a double ruled grating SPEX monochromator (1200 grooves/mm blazed at 330 nm). The emission beam was diffracted by a double ruled grating SPEX monochromator (1200 grooves/mm blazed at 500 nm). Singlet oxygen emission was detected through a double ruled grating SPEX monochromator (600 grooves/mm blazed at 1 µm) and a long-wave pass (780 nm). All spectra were measured in four-face quartz vials. All the emission spectra (fluorescence and singlet oxygen luminescence) have been displayed with the same absorbance (less than 0.2) with the lamp and photomultiplier correction.

Spectral overlap, as well as Förster radius, was computed to characterize the energy transfer from the Tb cation (Tb^3+^) to P1. Moreover, Tb luminescence decay profile was recorded using a Fluorolog spectrofluorometer; the excitation wavelength was set at 351 nm and the emission peaks were scanned in the 400–690 nm region. The luminescence lifetime of Tb alone or in mixture with P1 was recorded using lifetime Fluorolog. We assessed the 545 nm peak decay as it is the highest Tb fluorescence peak. If relevant, we computed the quenching constant (expressed as L.mol^−1^ s^−1^) as Kq = K/τ_0_, where K is the Stern-Volmer constant which was graphically determined; τ_0_ is the Tb fluorescence lifetime without photosensitizer.

### 4.7. Tb Scintillation Assessment under X-ray

Samples were irradiated using a biological X-ray Irradiator X-RAD 320 (Precision X-ray INC., North Branford, CT, USA) with a tungsten anode. Photons were produced by X-ray tubes and produced continuous energy distribution. The tube parameters were set from 25 up to 320 kVp (id est mean photon energies of 8 and 116 keV) with the current set from 0.5 up to 12.5 mA [59]. A 2 mm Al filter was used to remove low energy photons. For Tb scintillation assessment, irradiation time was set at 90 s for each parameter.

An optical fiber was inserted inside the irradiation chamber, in front of the vial containing AGuIX@Tb solutions to gather emission fluorescence photons. Emission spectra were recorded with an USB2000 spectrometer (Ocean Optics Inc, Dunedin, FL, USA). This versatile high-resolution spectrometer (FWHM = 3.5 nm) is an optical instrument based on a diffraction grating and a one-dimensional CCD detector array. Integration time was set to 5 s, the spectrum bandwidth ranged from 340 to 820 nm and the optical fiber was placed across from a transparent vial (UVette^®^ 220–1600 nm; cat.no. 952010051, Eppendorf, Hamburg, Germany). Emission spectra were recorded at different times to assess photonic density configurations on the Tb scintillation performance. Each measurement was repeated 7 times and all spectra were subtracted with the same solution spectrum obtained without irradiation. When varying X ray energy, we set the tube current to maximum and we set the voltage to 320 kVp when we assessed the tube current on the AGuIX@Tb response. A linear correlation coefficient was computed to characterize the relation between AGuIX@Tb peaks intensities, exciting photons energy (X ray kVp) and X ray flow (X ray mA), respectively. The energy transfer from the nanoparticles to a photo-agent was assessed with setting irradiation parameters to 320 kVp and 12.5 mA (a 3.0 Gy.min^−1^ dose rate in our experimental conditions).

### 4.8. Singlet Oxygen Production during Red Light Exposition or X-ray Irradiation

The reaction mixture was prepared in 30 mM Tris/HCl (pH 7.4) containing 400 µM AGuIX@Tb-P1 or 45 mM AGuIX@Tb and 5 µM APF or 10 µM SOSG probe. Singlet oxygen quenching was achieved by addition of sodium azide (NaN_3_; stock solution, 1 M) prepared in the same buffer, to a final concentration at 10 mM. Irradiation was set to 320 kVp, 12.5 mA, and source-surface distance adjusted to yield a 3.0 Gy min^−1^ dose rate. Fluorescence emission was detected spectroscopically at 515 and 525 nm for APF and SOSG, respectively. Home-made software allowed long acquisition times and synchronization between laser illumination and signal recording. Integration time was set to 100 ms and time points were acquired every 5 min during 20 min. Moreover, P1 at 100 µM was irradiated without a nanoscintillator with the same parameters to validate the absence of the direct excitation by X-rays.

### 4.9. Biological Experiments

#### 4.9.1. Cell Culture

Human U-251 MG (ECACC 09063001, Salisbury, UK) glioblastoma-derived cells were cultivated in Roswell Park Memorial Institute medium (RPMI) without phenol red, containing 10% (*v*/*v*) heat-inactivated (30 min at 56 °C) fetal calf serum (Invitrogen, Paisley, UK), 1% (*v*/*v*) non-essential amino acid (Invitrogen), 0.5% (*v*/*v*) essential amino acid (Invitrogen), 1 mM sodium pyruvate (Invitrogen), 1% (*v*/*v*) vitamin (Invitrogen) 0.1 mg.mL^−1^ of L-serine, 0.02 mg.mL^−1^ L-asparagine (Merck-Sigma), and 1% (*v*/*v*) antibiotics (10,000 U.mL^−1^ penicillin, 10 mg.mL^−1^ streptomycin) (Merck-Sigma). The cells were seeded routinely at 10^5^ cells mL^−1^ and cultivated at 37 °C in a 5% CO_2_ humidified atmosphere (Incubator Binder, Tübingen, Germany).

#### 4.9.2. Cell Growth Assessment

Impact of AGuIX@Tb or ZnPy3P1 on U-251 MG cell survival was assessed by the MTT procedure, based on the measurement of mitochondrial succinate dehydrogenase activity (EC 1.3.5.1) [60]. Cells were seeded in 96 well-plates at 10^4^ cells/well and left growing for 24 h. Cells were treated with increasing concentrations of ZnPy3P1 (up to 400 µM) and AGuIX@Tb (0.1 to 2 mM) over three days. In addition, glioblastoma cells were exposed to increasing concentrations of AGuIX@Tb-P1 or AGuIX@Gd-P1 (1 to 10 µM) for 72 h. At the time, the medium was discarded and replaced by 100 µL of complete medium containing 0.5 mg mL^−1^ MTT. The plates were incubated for 2 h at 37 °C and formazan crystals obtained were dissolved by adding 100 µL of DMSO. The plates were read at 540 nm (Multiskan Ascent spectrophotometer, Thermo Fisher Scientific, Illkirch, France). Results are expressed relative to those obtained from untreated cells (control), taken as 100. They represented quadruplicate determinations from two independent experiments (*n* = 8).

#### 4.9.3. Nanoparticles Cell Uptake

U-251 MG cells were seeded in 6 well-plates at 10^5^ cells/well and left to grow over 48 h. Cells were treated with increasing concentrations of nanoparticles (0.5 to 5.0 µM AGuIX@Tb-P1 or AGuIX@Gd-P1) over 48 h. After incubation, cell layers were washed with 2 mL of Dulbecco’s Phosphate buffer saline (DPBS, Merck-Sigma) and cells were suspended with 0.5 mL of 0.05% (*w*/*v*) Trypsin/0.02% (*w*/*v*) EDTA solution (Invitrogen) per well for 5 min at 37 °C. Complete medium (0.5 mL RPMI containing 10% (*v*/*v*) fetal calf serum) was then added. The cell suspension obtained was centrifuged at 1000 g for 5 min at 4 °C. Cell pellets were washed with 1 mL of DPBS and centrifuged again. Cells were suspended finally in 0.5 mL of DPBS and left on ice. Fluorescence of P1 was measured in 5000 cells/sample by flow cytometry (Gallios Analyzer, Beckman Coulter France, Roissy, France), with excitation/emission settings at 638 nm and 660/30 nm. Results obtained from nanoparticle uptake were expressed relative to those obtained reaching the maximum of nanoparticle absorption taken as 100. Results are expressed as mean ± SD from triplicate determinations from 3 independent experiments.

#### 4.9.4. PDT and X-PDT Conditions

Cell exposure to a red light was performed at 630 nm with a Laser diode (Biolitec, Biomedical Technology, Iena, Germany) at 0.7 W, corresponding to an irradiance at 4.54 mW.cm^−2^. Cell layers were exposed to 0.5 to 10.0 J cm^−2^, corresponding to an exposition duration of 2 min to 38 min. X-ray irradiation was performed using the X-ray Irradiator X-RAD 320 (Precision X-ray Inc., North Branford, CT, USA), using tube parameters set at 160 and 320 kV (mean photons energies of 58 and 116 keV) with current set at 10 mA. A 2 mm Al filter was used to remove low energy photons. The dose rates were 1.5 and 3.0 Gy min^−1^ for 160 and 320 kVp respectively. The doses delivered were 0.5, 2.0, 4.0, and 5.0 Gy for both dose rates.

#### 4.9.5. Reactive Species Quantification during PDT or X-PDT

U-251 MG cells were seeded in 6 well-plates at 2 × 10^5^ cells/well. Cells were left to grow over 48 h, then, treated in the presence of 1 µM AGuIX@-complexes for 24 h. Cells were washed with 2 mL of DPBS. Each well was filled again with complete medium before light exposition or X-ray irradiation. Reactive species generation measurements were achieved post treatment over 1 h. At each time point, cell medium was changed by 2 mL of pre-warmed medium containing 50 µM DCF2-DA for 30 min at 37 °C. Cells were then successively harvested by trypsination, washed with DPBS, and suspended in 0.5 mL DPBS before flow cytometry analysis. Reactive species generation was measured in 5000 cells/sample by flow cytometry with excitation/emission settings at 488 nm and 520/30 nm. Cell death was quantified also using propidium iodide in kinetic uptake experiments. Cells were treated with 20 µM propidium iodide (diluted in DMSO) added to the medium. Propidium iodide-positive cells were numbered by flow cytometric analysis with excitation/emission settings at 488 nm and 620/630 nm (FL3), respectively. Results represent the median of fluorescent peak. Results are expressed relative to those obtained from untreated cells, taken as 100. Results are expressed as mean ± SD of triplicate determinations from 3 independent experiments.

#### 4.9.6. Anchorage-Dependent Clonogenic Assay

The clonogenic assay was performed in 6 well-plates with U-251 MG cells seeded at 500 cells/well. Cells were then treated in the presence of 16.6 µM of AGuIX@Tb, 25 µM AGuIX@Gd, 1 µM AGuIX@Tb-P1 or 1 µM AGuIX@Gd-P1 (P1 equivalent concentration) for 24 h at 37 °C. After incubation, cells were washed with 2 mL of DPBS. 2 mL of complete medium were added in each well before X-ray radiation or red-light exposition. Cells were left to grow at 37 °C for 7 days. At time, cell clones were successively washed with 2 mL DPBS, fixed with 1 mL of 4% (*v*/*v*) formol (pH 7.4) at room temperature for 15 min, washed with 1 mL of DPBS, and stained for 30 min with 0.05% (*w*/*v*) crystal Violet solution prepared in DPBS and containing 25% (*v*/*v*) methanol. Finally, cells were washed three times with 2 mL of bi-distilled water. Cell clones obtained were analysed after picture capture (GelCount™, Oxford Optronix, Abingdon, UK) and ImageJ quantification (N.I.H., Bethesda, MA, USA). Image analysis was performed with well area taken as 862 mm^2^. Cell clone counting was improved by background subtraction. Data from untreated and treated cell conditions were compared and expressed as the mean ± SD (*n* = 12).

Survival fraction (SF) was calculated using the linear quadratic (LQ) model, based on the equation: SF_D_ = exp (−(αD + βD2)), where survival fraction is defined as SF_D_ = (plating efficiency at the dose D)/(platting efficiency at 0 Gy); D corresponds to the Gy dose; α and β, are determined from the established semi log curve, as SF_D_ = f (Gy dose). The effects of radiation alone or X-PDT, related to untreated cells or exposed to each AGuIX@NP, were compared by calculating the dose modifying factor (DMF) and the survival fraction at 2.0 Gy (SF_2_). We also determined the Dose Enhanced Ratio (DER) at 2.0 Gy. DER is defined as the ratio of SF_2.0 Gy_ calculated for untreated cells to that calculated of treated cells after irradiation (DER = SF_2.0 Gy_ (control cells)/SF_2.0 Gy_ (treated cells)). We finally calculated the enhanced factor, expressed in percentage, as EF (%) = 100 × (SF_2.0 Gy_ (control cells) − SF_2.0 Gy_ (treated cells))/SF_2.0 Gy_ (control cells)).

#### 4.9.7. Statistical Analysis

Results obtained were analyzed using the Kruskal-Wallis test (α = 0.05) and post-hoc Dunn’s test (α = 0.05) for paired groups. Any difference was considered significant at *p* < 0.05. Results obtained from clonogenic assays were analyzed using the Kruskal-Wallis test (with α = 0.05), and post-hoc by the Mann-Whitney test (α = 0.05) for unpaired groups.

## Data Availability

The datasets used and/or analysed during the current study are available from the corresponding author on reasonable request.

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
