# Peer review of "Terbium-Based AGuIX-Design Nanoparticle to Mediate X-ray-Induced Photodynamic Therapy"

_pharmaceuticals, 2021, doi:10.3390/ph14050396_

Round 1
Reviewer 1 Report
I revised the manuscript ID: pharmaceuticals-1176857 of Daouk et. al. entitled “Terbium-based AGuIX-design nanoparticle to mediate X-ray- 2
induced photodynamic therapy”
Briefly, the authors synthesized ultra-small nanoparticle chelated with Terbium as a nanoscintillator and porphyrin as a photosensitizer (AGuIX@Tb-P1) and was characterized for its photo-physical and physico-chemical properties. The effect of the nanoparticles was studied using human glioblastoma U-251 MG cells and was compared to treatment with AGuIX@ nanoparticles doped with Gadolinium and porphyrin (AguIX@Gd-P1). The manuscript is very well structured, very well written and the subject of the work is of great relevance for the treatment of cancer. After careful reading, I noticed some small errors that I pointed out throughout the pdf document. However, I was very surprised because the authors do not present the chapter on the conclusions of the work. Therefore, I suggest that this paper will be accepted after major revision.

Reviewer 2 Report
The manuscript under consideration discloses an X-ray induced photodyamic therapy based on replacement of Gd with Tb in AGuIX nanoparticle to enhance energy transfer to doped photosensitizer. With the metal replacement, the emission of the nanoparticle matched the absorption profile of the porphyrin photosensitizer that induces energy transfer to produce singlet oxygen. The authors then showed that their nanoparticle complex was able to be enriched in U-251 MG cells, and caused cytotoxic effect under red light illumination as well as X-ray irradiation. Considering the growing interest in X-PDT and the detailed characterization, it is this reviewer's opinion that this manuscript be published in Pharmaceuticals after minor revision. The following points may need to be addressed for improvement.
1. Section 2.6: red light exposure needs to be more accurately described in the main text. The author should consider describing mW.cm-2 and exposure time instead of just J.cm-2. Also what was the wavelength of the laser diode used in the experiment?
2. Figure 5c: how was the amount of "reactive species" quantified? The authors should mention the fluorescent probe, DCF2-DA used in the main text in both sections 2.7 and 2.6. By definition reactive (oxygen) species are so reactive that reacts with nearby biomolecules soon after their generation; as such it is more realistic to describe the measurement here as cellular oxidative stress (see Biochem J. 2010, 13, 428(2))
3. X-PDT is developed to operate without worrying about penetration depths. As such treatment of deep tumor in mouse model with red/NIR light or X-ray would greatly enhance the impact of the manuscript. A simplified version would be cell toxicity assay under X-ray/light radiation covered with a layer of meat to mimic tissue scattering.
4. The manuscript will be more digestible if it begins with a a graphical scheme describing the authors' nanoparticle particle design.
5. Typos in the text should be fixed. This includes but does not limit to line 244 Figure 5d -> Figure 5c, line 255 flux cytometry -> flow cytometry, Figure 6b missing concentration units
Author Response
Dear,
Find enclosed the response to Yours comments
Point 1: Section 2.6: red light exposure needs to be more accurately described in the main text. The author should consider describing mW.cm-2 and exposure time instead of just J.cm-2. Also what was the wavelength of the laser diode used in the experiment?
Response 1: We added precision of the conditions used in the main text as follows:
Lines 242-243: “U-251 MG cells were pre-treated with 1.0 and 2.5 µM AGuIX@Tb-P1 and exposed to a red light (630 nm, 0.7 W, irradiance at 4.54 mW.cm-²) to a fluence range of 2.5 to 10.0 J.cm-², corresponding to an exposition duration of 2 min to 38 min).”
- Figure 5c: how was the amount of "reactive species" quantified? The authors should mention the fluorescent probe, DCF2-DA used in the main text in both sections 2.7 and 2.6. By definition reactive (oxygen) species are so reactive that reacts with nearby biomolecules soon after their generation; as such it is more realistic to describe the measurement here as cellular oxidative stress (see Biochem J. 2010, 13, 428(2)).
Response 2: We make changes in section 2.6 and in section 2.7 (sentences added or sentences modified):
Line 221-224 Since NP absorption could lead to the increase of oxidative stress within the cells, we assessed whether nanoparticle uptake was related to reactive species generation, using DCF2-DA probe.
Line 250: “Since PDT is associated with the generation of oxidative stress…”
Line 253-255: “We used DCF2-DA probes which reacts with several reactive oxygen-derived species and gives a valuable estimation of the oxidative stress generated within the cells.”
Lines 265-266: “(c) Cells were pre-treated with 2.5 µM AGuIX@Tb-P1 (TbP1), followed by photodynamic treatment at 2.5 J.cm-² . Reactive species were quantified by DCF2-DA probe using flow cytometry.”.
Lines 275-277: We assessed whether oxidative stress was generated in AGuIX@Tb and AGuIX@Tb-P1 pre-treated cells and irradiated at 2.0 Gy, with an energy set at 320kVp, over 1 h post irradiation (Figure 6a), using DCF-2DA probe.
Line 284-286: (a) U-251 MG cells pre-treated with nanoparticles were exposed to X-ray irradiation at 2.0 Gy (320 kVp), then incubated with 50 µM DCF2-DA. The cells were harvested and reactive species were quantified by flow cytometry.
- X-PDT is developed to operate without worrying about penetration depths. As such treatment of deep tumor in mouse model with red/NIR light or X-ray would greatly enhance the impact of the manuscript. A simplified version would be cell toxicity assay under X-ray/light radiation covered with a layer of meat to mimic tissue scattering.
Response 3: The main objective of the present work was to characterise the new design of the AGuIX nanoparticle; specifically, whether the replacement of Gadolinium by Terbium will modify the behaviour of the nano-object; whether the nanoparticle could be used for X-PDT. Actually, we conduct in vivo experiments with U251 MG cells grafted in orthotopic site. The bio-distribution and toxicity of the nanoparticle will of course be investigated in the whole organism, in the tumour and in surrounding microenvironment. As discussed in the main text, our goal is also to limit X-ray deposit and to define conditions close to whose obtained in the experiments performed in vitro, which could limit cell damage observed in parenchymal cells surrounding the tumour in the clinic protocols (Calixto et al Molecules 2016, 21, 342).
- The manuscript will be more digestible if it begins with a graphical scheme describing the authors' nanoparticle particle design.
Response 4: We added a graphical scheme of the nanoparticle design (Scheme 1 and legend, lines 124-129)
- Typos in the text should be fixed. This includes but does not limit to line 244 Figure 5d -> Figure 5c, line 255 flux cytometry -> flow cytometry, Figure 6b missing concentration units
Response 5: We edited the text again. In addition, we change the text in the legend of figure 6b (lines 288-290) to clarify the experiments performed, hopping that You will agree with this proposal.
Reviewer 3 Report
This is an interesting manuscript that describes a novel approach to X-ray induced PDT using porphyrin-conjugated Tb rather than Gd nanoparticles. The work has been carried out competently and there are no obvious flaws in the methodology so it can be published after minor revisions. My only significant criticism would be that the quality of the English is not at the level it should be. The manuscript would definitely benefit from being proofread and corrected by a native English speaker and I recommend that this is done at this point. I would urge the authors to also consider the following points when preparing their revised manuscript:
(a) Should the structure of P1 not be more clearly defined in the text and abstract with a Chemdraw graphic provided as a figure in the introduction in the context of the functionalized nanoparticles? The abbreviation P1 is used repeatedly before mention is made of it being (5-(4-carboxyphenyl succinimide ester)-10,15,20-triphenylporphyrin on line 105.
(b) Is the use of three significant figures for data values in Table 1 justifiable?
(c) Decimal points should be used for the R2 values In Figure 2(c) as is the case in Figure 2(d).
(d) There are several minor formatting issues that should be fixed after a careful reread. The strikethrough as part of "ZnPy3P1" on line 559 is particularly noteworthy in this context and suggests the authors should have taken greater care in this regard prior to submission.
(e) Should ® not be superscripted throughout?
(f) A multiplication sign should be used where appropriate rather than the letter "x", e.g. line 633.
(g) There should be no space after a minus sign for a negative number, e.g. line 644.
Author Response
Dear,
Find enclosed the response to Yours comments
This is an interesting manuscript that describes a novel approach to X-ray induced PDT using porphyrin-conjugated Tb rather than Gd nanoparticles. The work has been carried out competently and there are no obvious flaws in the methodology so it can be published after minor revisions.
English: My only significant criticism would be that the quality of the English is not at the level it should be. The manuscript would definitely benefit from being proofread and corrected by a native English speaker and I recommend that this is done at this point. I would urge the authors to also consider the following points when preparing their revised manuscript:
English: The text has been edited again and was proofread by Pr Simon Thornton, native from New Zealand.
Point 1 (a): Should the structure of P1 not be more clearly defined in the text and abstract with a Chemdraw graphic provided as a figure in the introduction in the context of the functionalized nanoparticles? The abbreviation P1 is used repeatedly before mention is made of it being (5-(4-carboxyphenyl succinimide ester)-10,15,20-triphenylporphyrin on line 105.
Response 1: We added a graphical scheme of the nanoparticle design. The full name of the abbreviation P1 has been added in the synopsis and after first appearance in the main text.
(b) Is the use of three significant figures for data values in Table 1 justifiable?
Response 2 (b). We suppose that You refer to figure 1. We firstly drew the figures 1a and 1b (and also figure 2a and 2b) with a 8cm-width and 8cm-high. However, the integrated lines from each measurement were so close that it was difficult to discriminate them from each other.
(c) Decimal points should be used for the R2 values In Figure 2(c) as is the case in Figure 2(d).
Response 3 (c): the figure 2c has been designed again and corrected
(d) There are several minor formatting issues that should be fixed after a careful reread. The strikethrough as part of "ZnPy3P1" on line 559 is particularly noteworthy in this context and suggests the authors should have taken greater care in this regard prior to submission.
Response 4: (d) We have edited the main text again. We found other typos errors that have been corrected.
(e) Should ® not be superscripted throughout?
Response 5 (e): Corrections have been made
(f) A multiplication sign should be used where appropriate rather than the letter "x", e.g. line 633.
Response 6 (f): the sign has been added
(g) There should be no space after a minus sign for a negative number, e.g. line 644.
Response 7(g): Corrected
Round 2
Reviewer 1 Report
I revised the manuscript ID: pharmaceuticals-1176857 of Daouk et. al. entitled “Terbium-based AGuIX-design nanoparticle to mediate X-ray- 2 induced photodynamic therapy” after the authors made the changes suggested by me and by other reviewers on the first submission. The paper has been improved with our suggestions therefore, I recommend accepting this manuscript.